# Experimental determination and mathematical modeling of standard shapes of forming autophagosomes

Yuji Sakai [1,2,3] ✉, Satoru Takahashi[1,4], Ikuko Koyama-Honda [1], Chieko Saito[1] & Noboru Mizushima [1] ✉

The formation of autophagosomes involves dynamic morphological changes of a phagophore from a flat membrane cisterna into a cup-shaped intermediate and a spherical autophagosome. However, the physical mechanism behind these morphological changes remains elusive. Here, we determine the average shapes of phagophores by statistically investigating three-dimensional electron micrographs of more than 100 phagophores. The results show that the cup-shaped structures adopt a characteristic morphology; they are longitudinally elongated, and the rim is catenoidal with an outwardly recurved shape. To understand these characteristic shapes, we establish a theoretical model of the shape of entire phagophores. The model quantitatively reproduces the average morphology and reveals that the characteristic shape of phagophores is primarily determined by the relative size of the open rim to the total surface area. These results suggest that the seemingly complex morphological changes during autophagosome formation follow a stable path determined by elastic bending energy minimization.

Eukaryotic cells have membranous organelles with a variety of characteristic shapes. The autophagosome, which mediates macroautophagy (hereafter simply referred to as autophagy), is unique among the various organelles in that its generation involves characteristic membrane deformation. Autophagy is a bulk degradation process, through which a portion of the cytoplasm is sequestered into autophagosomes and is degraded upon fusion with lysosomes (Fig. 1)[1]. In this process, a flat cisterna called a phagophore (also called an isolation membrane) grows into a spherical stomatocyte through a cup-shaped intermediate. After the closure of the edge, it becomes a double-membrane autophagosome. Despite these characteristic morphological changes, there have been no systematic and quantitative studies to date on the standard morphology of phagophores during autophagosome formation. The morphology of phagophores is often approximated as a simple geometric shape, such as a part of a sphere or an ellipsoid, but the complexity and diversity of their morphology remain largely unexplored.

Quantitative and theoretical analyses can elucidate the mechanisms that control organelle morphology. Quantitative comparisons between experimentally observed and theoretically calculated shapes provide a mesoscopic, molecular-level explanation of the mechanisms that control organelle morphology[2–4]. In theoretical analyses, organelles are often considered to be simple geometric shapes, such as spheres or tubes, and their complex and diverse morphologies, and thus the mechanisms behind them, are often ignored. Although theoretical research on autophagy is only in its early phase, the morphology of phagophores has been analyzed by approximating parts of a sphere or an ellipse[5,6]. However, the actual morphology of phagophores observed in cells is more complex, and previous studies have oversimplified their morphology.

[1]Department of Biochemistry and Molecular Biology, Graduate School of Medicine, The University of Tokyo, Bunkyo-ku, Tokyo 113-0033, Japan. [2]Department of Biosystems Science, Institute for Life and Medical Sciences, Kyoto University, Sakyo-ku, Kyoto 606-8507, Japan. [3]Interdisciplinary Theoretical and Mathematical Sciences (iTHEMS) Program, RIKEN, Wako, Saitama 351-0198, Japan. [4]Department of Neurosurgery, Graduate School of Medical and Dental Sciences, Tokyo Medical and Dental University, Bunkyo-ku, Tokyo 113-8510, Japan. ✉e-mail: yuji-sakai@m.u-tokyo.ac.jp; nmizu@m.u-tokyo.ac.jp

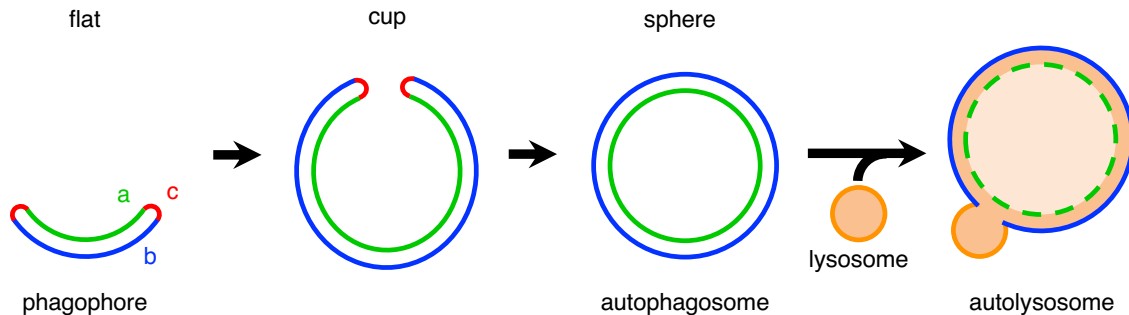

**Fig. 1 | Schematic representation of autophagosome formation.** Each shape represents an axially symmetric membrane structure. As surface area increases, the shape of a forming autophagosome (phagophore) changes from an initial flat cisterna, through a cup-shaped intermediate, to a spherical closed autophagosome. The regions labeled a, b, and c correspond to the inner membrane, the outer membrane, and the rim, respectively. Eventually, autophagosomes and lysosomes fuse to degrade the contents of the former.

In the present study, we statistically and quantitatively investigated the morphology of phagophores. First, we determined the average morphology of more than 100 phagophores obtained from three-dimensional (3D) electron micrographs. The results showed that phagophores were elongated vertically, with a rim curved outward to form a catenoid-like shape. To understand the morphological characteristics, we developed a theoretical model based on the elastic bending energy. The morphology of the phagophore was considered to be in equilibrium at each step[6]. The physical parameter, Gaussian modulus, which determines the elastic properties of the phagophore membrane, was estimated from the experimental shape of the rim. The resulting theoretical model quantitatively reproduced the morphology during autophagosome formation observed by electron microscopy. These results suggest that autophagosomal membranes are highly flexible and that the characteristic shapes of phagophores are primarily determined by elastic bending energy minimization.

## Results

### Quantification of phagophore morphology from 3D-electron micrographs

To understand the standard shapes of phagophores, we determined the average morphology of phagophores in mouse embryonic fibroblasts by array tomography, a technique that can be used in 3D electron microscopy[7]. Images of more than 100–200 serial sections (25-nm thickness) of a region containing several cells were captured using a scanning electron microscope[8]. We collected 117 entire 3D structures of phagophores from 23 chemically fixed cells based on their characteristic 3D morphology and strong contrast after osmium staining[9] (Fig. 2a). Next, each phagophore membrane was segmented and approximated as a point cloud $\{\mathbf{X}_{i,j}\}$ of 300-600 points, where each point $\mathbf{X}_{i,j}$ represented the location of the $j$-th point on the membrane of the $i$-th phagophore structure (Fig. 2b). In this electron microscopy experiment, cells were fixed with reduced osmium tetroxide to keep the inner and outer membranes attached to each other, which is their natural physiological state; if conventional osmium tetroxide had been used, the space between the outer and inner membranes would have been artificially expanded[10]. Thus, the inner and outer membranes were treated as a single membrane in each point cloud representation. Each 3D morphology was constructed by stacking a set of points obtained from a series of images (Fig. 2c).

To obtain an average shape of structures with different sizes, we normalized the size of each 3D phagophore using standard $z$-score normalization since the morphologies were almost size-independent (Supplementary Fig. 1). In $z$-score normalization, each point $\mathbf{X}_{i,j}$ was converted to a standardized point $\mathbf{Y}_{i,j}$ as

$$\mathbf{Y}_{i,j} = \frac{\mathbf{X}_{i,j} - \boldsymbol{\mu}_{X_i}}{\sigma_{X_i}}, \boldsymbol{\mu}_{X_i} = \frac{1}{N_i}\sum_{j=1}^{N_i}\mathbf{X}_{i,j}, \sigma_{X_i} = \sqrt{\frac{1}{N_i}\sum_{j=1}^{N_i}\left|\mathbf{X}_{i,j} - \boldsymbol{\mu}_{X_i}\right|^2}, \quad (1)$$

where $\boldsymbol{\mu}_{X_i}$, $\sigma_{X_i}$, and $N_i$ represent the mean, deviation, and number of the points belonging to the $i$-th phagophore, respectively. $\sigma_{X_i}$ means the structure size. Then, the normalized point cloud $\{\mathbf{Y}_{i,j}\}$ was obtained.

Because phagophores in the electron microscopy images were oriented in various directions, we rotated each structure to be oriented in a uniform direction (along the $z$-axis). The average edge vector of a normalized point cloud, $\boldsymbol{\mu}_{e_i}$, was used as the orientation of the $i$-th phagophore and was obtained as

$$\boldsymbol{\mu}_{e_i} = \frac{1}{M_i}\sum_{j \in edge}^{M_i}\mathbf{Y}_{i,j}, \quad (2)$$

where the edge points belonging to the cloud were summed and $M_i$ was the number of edge points. From the average edge vector, the polar angle, $\theta_{e_i}$, and azimuth angle, $\varphi_{e_i}$, of the structure were determined. Each point $\mathbf{Y}_{i,j}$ was rotated according to

$$\mathbf{Z}_{i,j} = R_y(-\theta_{e_i})R_z(-\varphi_{e_i})\mathbf{Y}_{i,j}, \quad (3)$$

where $R_y$ and $R_z$ are the matrices that rotate $\mathbf{Y}_{i,j}$ by $-\theta_{e_i}$ about the $y$-axis and $-\varphi_{e_i}$ about the $z$-axis, respectively, and then, a point cloud with uniform size and orientation, $\{\mathbf{Z}_{i,j}\}$, was obtained. The surface was constructed from the normalized point cloud by Delaunay triangulation (Fig. 2d and Supplementary Movie 1). Thus, we reconstructed the normalized shapes of phagophores obtained from the 3D electron micrographs.

Because the obtained shapes of phagophores were diverse (reflecting the morphological changes during autophagosome formation), we first classified them into four categories using unsupervised machine learning clustering with the k-means method. The resulting four categories appeared to be primarily characterized by differences in rim size (Supplementary Fig. 2). Thus, we selected four representative categories based on the rim size. The rim size $\sigma_{e_i}$ was expressed as the magnitude of the deviation of the normalized point cloud belonging to the rim as

$$\sigma_{e_i} = \sqrt{\frac{1}{M_i}\sum_{j \in edge}^{M_i}\left|\mathbf{Z}_{i,j} - \boldsymbol{\mu}_{e_i}\right|^2}. \quad (4)$$

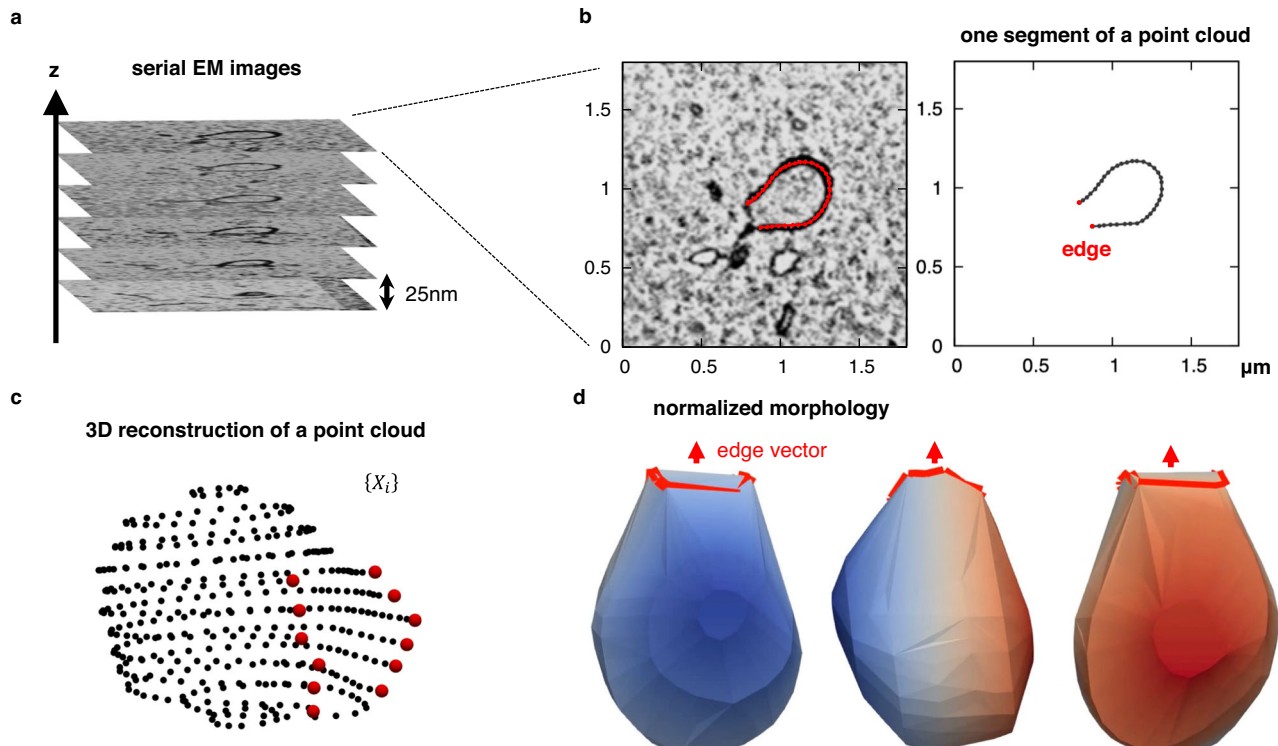

**Fig. 2 | Three-dimensional electron microscopy of phagophores. a** Serial electron microscopy (EM) images of a phagophore. **b** Left, an electron microscopy section image; the red dots represent the contour of the phagophore. Right, one segment of a point cloud of the extracted contour; the two red dots represent the edges of the phagophore. Serial section images of 117 autophagosomes were randomly selected for analysis from two resin-embedded blocks (6 and 17 cells from each block). **c** 3D reconstruction of a point cloud. **d** A normalized shape of a phagophore, in which the deviation of the point cloud was normalized to one and rotated so that the direction of the edge aligns with the *z*-axis. Each structure is shown rotated 90 degrees about the *z*-axis. Rotating views (the *z*-axis is the rotation axis) are shown in Supplementary Movie 1. The surface was constructed by Delaunay triangulation, in which the open edge of the cup (the area surrounded by the red line) was also considered as part of the surface. Source data are provided as a Source Data file.

The four categories of observed shapes were as follows: very early-cup ($\sigma_{e_i} > 1.1$), early-cup ($0.8 < \sigma_{e_i} < 1.0$), middle-cup ($0.5 < \sigma_{e_i} < 0.7$), and late-cup ($\sigma_{e_i} < 0.4$), with 15, 16, 24, and 45 observations, respectively (examples of 3D electron micrographs of each category are shown in Supplementary Figs. 1–3). The point clouds in each category were superposed (Fig. 3a–d and Supplementary Movies 2–5). The spread of the point clouds indicated the variation within each morphology, while the regions with a high density of points represented the modal morphology. The superimposed point clouds showed less variation and appeared to be axisymmetric about the *z*-axis. This was also true for individual phagophores, although they were slightly deformed by intracellular fluctuations (Supplementary Figs. 1 and 4).

Thus, these point clouds were projected onto the radial and axial planes (Fig. 3e–h). The average morphology obtained suggested that the cup-shaped phagophores had two characteristic features. First, they were elongated vertically along the rotation axis. This result was consistent with our previous observations by fluorescence microscopy[11]. Second, as the phagophore membrane approached the rim, it curved inward and then outward, forming a catenoid shape. The outward curvature was higher in late-cups. However, the rim appeared to be straight and not curved outward in very early-cups. These characteristic features were also observed in individual phagophores in electron micrographs (Fig. 3i–l).

One methodological concern was that these characteristic shapes might have resulted from the chemical fixation of cells during sample preparation for electron microscopy. To rule out this possibility, we prepared samples by the high-pressure freezing and freeze-substitution method and subjected them to electron microscopy. The results showed similar morphological features of phagophores, such as elongated cups with catenoid-like rims (Supplementary Fig. 4). Furthermore, we directly observed phagophores in live cells expressing GFP-LC3, a marker for phagophores and autophagosomes. Typical shapes of elongated phagophores with a catenoidal rim, which were consistent with the superimposed electron microscopy data, were frequently observed (Fig. 3m–p and Supplementary Fig. 5). Thus, the structures observed by electron microscopy should represent the actual morphology of those in live cells.

To determine the average shape and quantify the curvature of the membrane, the projected point clouds were fitted with the *z*-axis symmetric polynomial function

$$x^2 = az^3 + bz^2 + cz + d, \tag{5}$$

where the fitting parameters are summarized in Supplementary Table 1 (Fig. 4a–d). The standard error of each parameter in the early-, middle- and late-cup shapes was within a few percentage points, while the error of the very early-cup shape was larger because of the small number of statistics and the large classification range ($\sigma_e > 1.1$). Each fitting curve well reproduced the point clouds that were experimentally obtained by electron microscopy. The fitted curves of middle- and late-cups clearly had a catenoidal rim (curved outward). The 3D surface in the axisymmetric approximation was represented using the fitting function

$$(f(v)\cos u, f(v)\sin u, v), \ f(v) = \sqrt{av^3 + bv^2 + cv + d}, \tag{6}$$

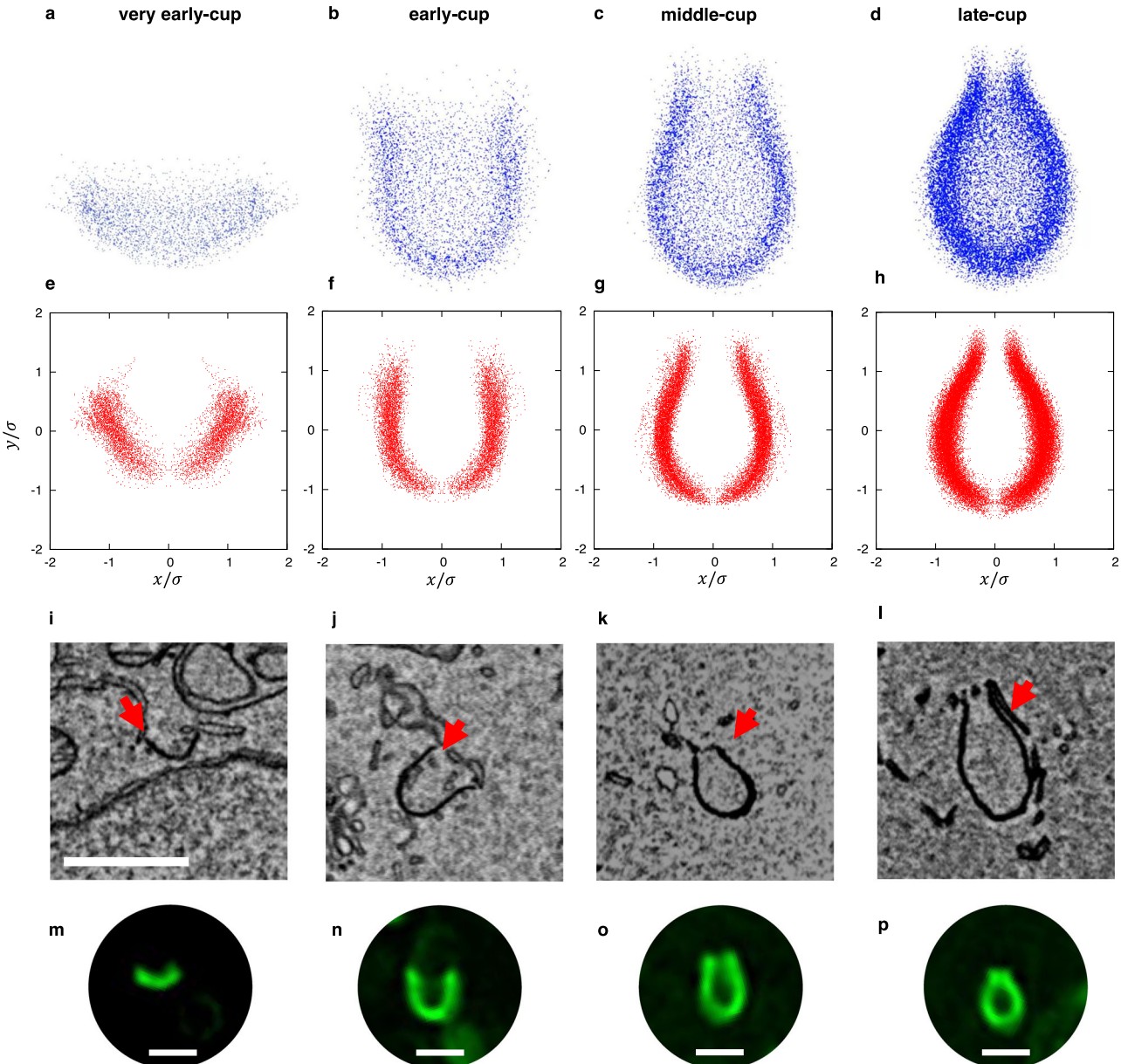

**Fig. 3 | Average shape of phagophores. a–d** Three-dimensional point clouds of very early-cups (**a**), early-cups (**b**), middle-cups (**c**), and late-cups (**d**) are shown. Point clouds were obtained from electron microscopy images of 15 very early-cups (3491 points), 16 early-cups (4585 points), 24 middle-cups (8578 points), and 45 late-cups (17,692 points) and then superposed. Rotating views (the *z*-axis is the rotation axis) are shown in Supplementary Movies 2–5. **e–h** The projected point clouds of very early-cups (**e**), early-cups (**f**), middle-cups (**g**), and late-cups (**h**) onto the radial (*x*) and axial (*y*) plane. The unit of length was normalized so that the variance of the point set was one. (**i–l**) Examples of electron microscopy images of phagophores at the very early-cup (**i**), early-cup (**j**), middle-cup (**k**), and late-cup (**l**) stages. The red arrows indicate phagophores; scale bar, 1 μm. **m–p** Examples of live-cell imaging of GFP-LC3 showing phagophores at the very early-cup (**m**), early-cup (**n**), middle-cup (**o**), and late-cup (**p**) stages; scale bar, 1 μm. More than 30 cup-shaped autophagosomes from two biological replicates were observed by 3D time-lapse microscopy and they showed consistent results. Source data are provided as a Source Data file.

where *u* represents the azimuth angle. The total curvature *J* (indicated by the line color in Fig. 4a–d) on the surface at $z = v$ was obtained by

$$J(v) = J_m + J_p, \quad J_m = -\frac{f''(v)}{\left[(f'(v))^2 + 1\right]^{\frac{3}{2}}}, \quad J_p = \frac{1}{f(v)\left[(f'(v))^2 + 1\right]^{\frac{1}{2}}}, \quad (7)$$

where $J_m$ and $J_p$ represent the meridional (i.e., perpendicular) and parallel curvature relative to the *xy*-plane, respectively (Fig. 4e, line colors in Fig. 4f–i). In middle- and late-cups, the parallel curvature, $J_p$, increased, with a positive sign as the surface approached the rim from the bottom, while the meridional curvature, $J_m$, decreased, with a

negative sign. Furthermore, the meridional curvature decreased with the rim size. This trend was also observed in individual phagophores (Fig. 4j), where the rim curvature was obtained by fitting the point cloud of each structure with Eq. (6). Thus, the two orthogonal curvatures $J_m$ and $J_p$ had opposite signs and seemed to cancel each other out, thus reducing the total curvature *J*.

## Modeling of phagophore morphology

Next, to investigate whether the morphological features of the phagophores obtained by electron microscopy at each stage could be spontaneously determined primarily according to the elastic model based on the physical properties of the membrane, we conducted a

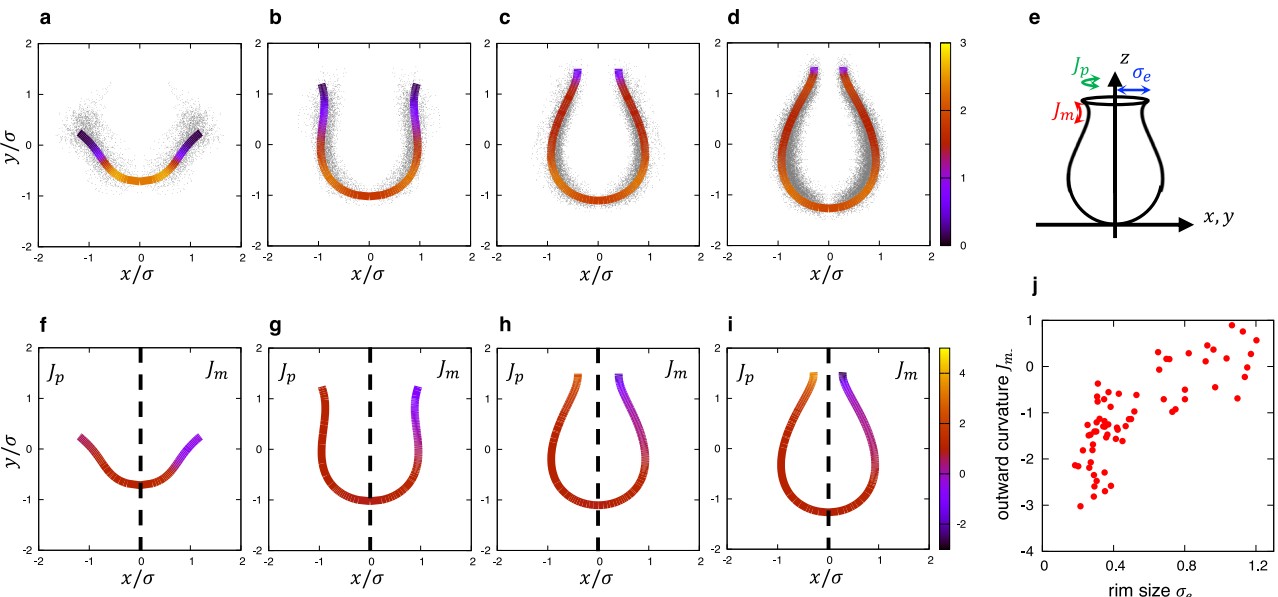

**Fig. 4 | Shapes and curvatures fitted with approximate curves. a–d** The shapes fitted with polynomial functions of a very early-cup (**a**), an early-cup (**b**), a middle-cup (**c**), and a late-cup (**d**), in which the line color represents the total curvature. In each panel, the black dots represent the same point set as in Fig. 3e–h, respectively. The unit of length was normalized so that the variance of the point set was one. **e** A schematic description of the meridional and parallel curvatures relative to the *xy*-plane, $J_m$ and $J_p$, respectively. **f–i** The meridional ($J_m$, $x \geq 0$, right) and parallel ($J_p$, $x \leq 0$, left) curvatures of a very early-cup (**f**), an early-cup (**g**), a middle-cup (**h**), and a late-cup (**i**) are shown by the various line colors; the shape of the fitting curve in each panel is the same as those in panels **a–d**. **j** A scatter plot of the meridional outward curvature near the rim obtained by polynomial fitting of individual phagophores. Source data are provided as a Source Data file.

mathematical model analysis. Each shape of the phagophore membrane shown in Fig. 3 (i.e., very early-, early-, middle-, and late-cups) is stable for tens of seconds to a minute during autophagosome formation[6,11]. While the rim length changes on this time scale, $\tau_{grow} \sim 1\,min$, the mechanical relaxation time of the membrane is $\tau_{relax} \sim 1\,ms$[6]. Because $\tau_{relax} \ll \tau_{grow}$, it was considered that the rim length remained constant during the mechanical relaxation and each morphology observed in electron microscopy was in equilibrium with a fixed rim length. The membrane shapes at equilibrium were determined from the elastic bending energy model. In the elastic bending energy model[12], the free energy $F$ for a fixed rim length was given by

$$F = \int \left( \frac{\kappa_b}{2} (J - J_0)^2 + \kappa_G K \right) dA + \gamma_A A + \gamma_L L. \qquad (8)$$

Here, the first term is the elastic bending energy with the total curvature, $J = J_m + J_p$, and Gaussian curvature, $K = J_m J_p$, of the inner and outer membranes, which is scale-invariant. The spontaneous curvature, $J_0$, of the inner and outer membranes could be caused by asymmetries in the composition of proteins and lipids between the inner and outer membranes. It is considered that the phagophore membranes expand by acquiring lipids from the endoplasmic reticulum through the lipid transfer protein ATG2[13–15], thereby keeping the lumen (i.e., intermembrane space) volume small and the inner and outer membranes almost parallel (Fig. 3i–l). Thus, the phagophore was assumed to have closely juxtaposed inner and outer membranes with the same curvature and area, and the rim was approximated as a closed line (Fig. 5a). The bending modulus is $\kappa_b = 20k_B T$ with the Boltzmann constant $k_B$ and temperature $T$. The Gaussian modulus of elasticity, which was estimated from the experimental morphology using the force balancing conditions acting on the rim at equilibrium, $\kappa_G = -0.2\kappa_b$, was used (Fig. 5b). The smaller the Gauss modulus, the more the rim curves outward (Fig. 5c–f). The second and third terms are the constraints for the membrane area $A$ and the rim length $L$, respectively, while $\gamma_A$ and $\gamma_L$ are the Lagrange multipliers for fixing $A$ and $L$, respectively. The second term is the rim line energy, which

corresponds to the elastic bending energy of the rim. The rim length is in principle controlled by the abundance of curvature-inducing proteins that stabilize a highly curved rim[6]. The line energy includes the effects of the spontaneous curvature induced by these proteins (see Methods). For an axisymmetric membrane, the free energy was rewritten as

$$F = 2\pi \int_0^{s_1} \left[ \frac{\kappa_b}{2} x \left( J_m + J_p - J_0 \right)^2 + \kappa_G x J_m J_p + \gamma_A x + \gamma_L \dot{x} \right] ds, \qquad (9)$$

with

$$J_m = \dot{\theta}, \quad J_p = \frac{\sin \theta}{x}, \qquad (10)$$

where $s$, $s_1$ $x$, $\theta$, and $J_0$ are the arc length, location of the rim, radial coordinate, tilt angle, and the spontaneous curvature, respectively (Fig. 5a). The dots represented the derivative with respect to $s$. The membrane morphology was obtained by the variational method[16]. A complete method for obtaining the morphology from the bending energy is described in the Methods.

First, we determined the morphology of a stable cup-shaped membrane with several different rim radii in the absence of spontaneous curvature (Fig. 6a–d). Because the free energy and the stable membrane morphology are scale-invariant, the rim radius was rescaled as

$$l = x_r / \sqrt{A/2\pi}, \qquad (11)$$

with membrane area $A = 4\pi \int_0^{s_1} x\,ds$ and rim radius $x_r = \int_0^{s_1} \cos \theta\,ds$, where $l = 1$ represents a flat cisterna shape and $l = 0$ a closed sphere shape. Cups for which $0.2 < l < 0.6$ were not spherical but elongated vertically along the rotational axis. This trend was also observed in our previous models[6]. In these structures, the curvature in the elongated direction was reduced, thus lowering the bending energy (a cylinder [radius $r$, height $h$] had less bending energy than a spherical cap [radius

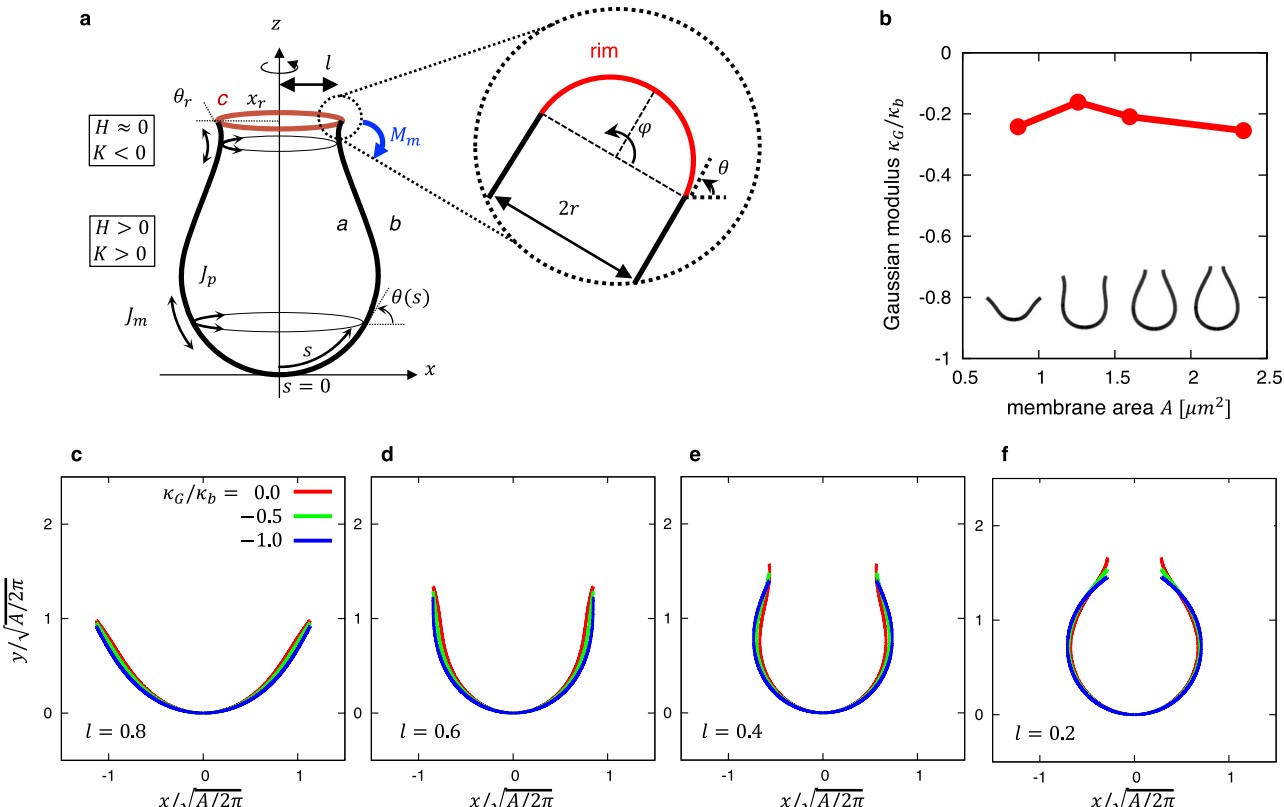

**Fig. 5 | Model description of phagophore and Gaussian modulus. a** A schematic description of the phagophore. *a* and *b* represent the inner and outer membranes, respectively, which are approximated by two paralleled axisymmetric membranes. *c* represents the rim which is approximated as a line. *z* is the axis of symmetry, *s* is the length along the contour measured from the origin, and $\theta$ is the angle between the tangent to the contour and the *x*-axis. $M_m$ is the meridional moments. $J_m$ and $J_p$ are the two principal curvatures. *J* and *K* are the total and Gaussian curvatures, respectively, with $J > 0$, $K > 0$ for convex surfaces and $J \approx 0$, $K < 0$ for catenoidal surfaces. **b** Gaussian modulus obtained by fitting the experimental shapes, where $\kappa_b = 20 k_B T$. The membrane area, $A = 4\pi \int f \sqrt{1 + (f')^2} dx$, is also obtained from the polynomial fitting. **c–f** Membrane shapes obtained from the bending energy with different values of the Gaussian modulus, $\kappa_G = 0$ (red), $-0.5\kappa_b$ (green), and $-\kappa_b$ (blue), for different values of the rim radius, $l = 0.8$ (**c**), $l = 0.6$ (**d**), $l = 0.4$ (**e**), and $l = 0.2$ (**f**). The unit of length was non-dimensionalized by the length, $\sqrt{A/2\pi}$. Source data are provided as a Source Data file.

*r*] of the same area when *r*>*h*). Spontaneous curvature made the membrane morphology more spherical (Supplementary Fig. 6). Our experimental results indicating that phagophores adopt elongated cup-shaped morphologies suggest that the spontaneous curvature of the membrane is not high. Furthermore, these structures had a catenoid-like rim. The catenoidal rims were attributed to the fact that the Gaussian modulus values of the phagophores were not particularly small. As the Gaussian modulus became even lower, the catenoid-like rim disappeared, and the theoretical model could no longer reproduce the experimental results (Fig. 5c–f). As the rim radius became smaller, the warping became stronger. These morphological features obtained by the theoretical model were consistent with those obtained by electron microscopy (Fig. 3). As phagophore morphology changed, the elastic properties of the membrane also changed (Fig. 6a–d). In our model, the equilibrium shape depends on the membrane elastic moduli ($\gamma_A, \gamma_L, \kappa_G$). The rim radius *l* increases with decreasing $\gamma_A$, and $\kappa_G$ increases with $\gamma_L$ (Supplementary Fig. 7). Therefore, as the membrane grows, changes in the lipid and protein composition of the phagophore membrane would modulate the physical properties of the membrane. Note that $\gamma_A$ was always negative (Fig. 6a–d), which indicates that the surface energy, the second term of Eq. (8), was negative, and thus, membrane area growth decreased the energy of cup shapes.

Next, we calculated the total curvature to quantitatively evaluate the shapes obtained by the theoretical model (Fig. 6a–d). In all cup-shaped structures, the total curvature was reduced near the rim. The meridional curvature, $J_m$, and parallel curvature, $J_p$, were also calculated (Fig. 6e–h). The meridional curvature was negative near the rim, reflecting the outward curvature of the membrane, while the parallel curvature was positive. As a result of the two competing curvatures, negative and positive, the total curvature near the rim seemed to be smaller. Except for the rim, which was isotropic, the two curvatures were of approximately the same magnitude.

We further investigated the change of curvatures during the morphological transition (Fig. 6i). As the rim radius decreased, the parallel curvature $J_p$ at the rim increased monotonically because the curvature given by $J_p = \sin\theta/x$ was approximately inversely proportional to the radial distance *x*. In the middle- and late-cup shapes, the surface near the rim got closer to the rotational axis, and therefore the parallel curvature, $J_p$, increased. On the other hand, as the rim radius decreased, the meridional curvature, $J_m$, decreased (became more negative) in opposition to the parallel curvature, $J_p$, keeping the total curvature ($J = J_m + J_p$) small. Thus, the two orthogonal curvatures canceled each other out, making the rim catenoidal.

## Comparison between theoretical and experimental morphologies

We compared the results of our theoretical calculations with our experimental data. The point clouds obtained from electron micrographs (Fig. 3e–h) were superimposed on the theoretically calculated morphologies with the same rim sizes (Fig. 7). At each step, the experimental and theoretical results overlapped well. In particular, the

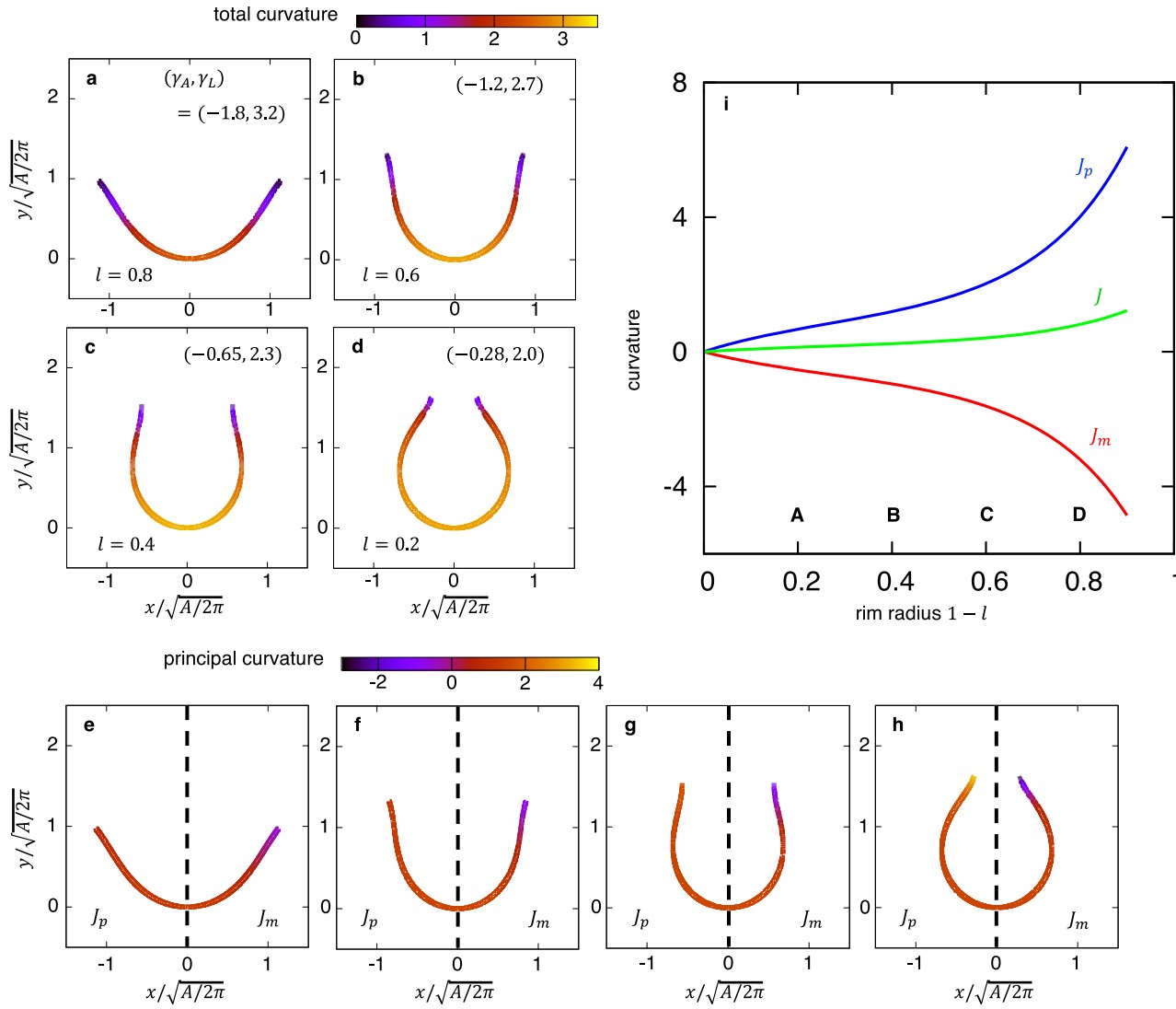

**Fig. 6 | Shapes and curvatures obtained from theoretical calculations.**
**a–d** Membrane shapes obtained from the bending energy with Gaussian modulus $\kappa_G = -0.2\kappa_b$ for several values of the rim radius $l = 0.8$ (**a**), $l = 0.6$ (**b**), $l = 0.4$ (**c**), and $l = 0.2$ (**d**), where the line color represents the total curvature. **e–h** The meridional and parallel curvatures $J_m$ and $J_p$ are shown in the right ($x \ge 0$) and left ($x \le 0$) regions, respectively. The same shapes as Panel **a–d** were used. **i** The meridional, parallel, and total curvatures, $J_m$ (red), $J_p$ (blue), and $J$ (green), respectively, at the rim as a function of the rim radius, $l$. The rim radii corresponding to Panel **a–d** are indicated. The unit of length was non-dimensionalized by the length $\sqrt{A/2\pi}$. Source data are provided as a Source Data file.

longitudinally elongated shape and the catenoidal rim were well captured by the theoretical calculations. The individual structures of various sizes at each step were also compared with the theoretical morphology (Supplementary Fig. 1). Over a wide range of size scales, the theoretical model reproduced the experimentally observed morphology well. The relative error of the theoretical curves to the experimental fitting curves was within 30% across all shapes (Supplementary Fig. 8). The magnitude of this error was $\sim k_B T$ in energy, which was within the range of thermal fluctuations. These results suggest that the morphological transition of phagophores during autophagosome formation follows a stable path determined by the elastic bending energy.

However, there was a slight discrepancy between the theoretical expectation and experimental data, especially at the late-cup stage (Fig. 7). This discrepancy may be partially explained by non-equilibrium fluctuations. The outer and rim membranes of the phagophore often attach to the endoplasmic reticulum[10,17,18], and the phagophore membrane may actively fluctuate like the endoplasmic reticulum membrane. Cytoskeletons may also contribute to autophagosome shaping[19,20]. Another possibility is the influence of lipid asymmetry. During autophagosome formation, lipids are provided to the outer leaflet by the lipid transfer protein ATG2[13–15]. Although it is hypothesized that the autophagosomal scramblase ATG9 resolves lipid asymmetry[21], the remaining asymmetry may contribute to the elongation of phagophores[22]. However, these effects on the shape of the phagophore would be within the range of thermal fluctuation.

## Discussion

The shape of phagophores has been empirically assumed, but their standard shapes have not yet been determined experimentally. In this study, we quantitatively analyzed the average morphology of phagophores using 3D electron microscopy data and found that phagophores are vertically elongated and that their rims have a catenoidal shape. These morphological features are also observed in cryo-electron tomography images of yeast phagophores[23]. We also showed that these characteristic features could be well explained by an elastic bending energy model. These results suggest that phagophore membranes are highly flexible and the morphological changes during autophagosome formation follow a stable path determined by elastic bending energy minimization. Our results also suggest that the shape

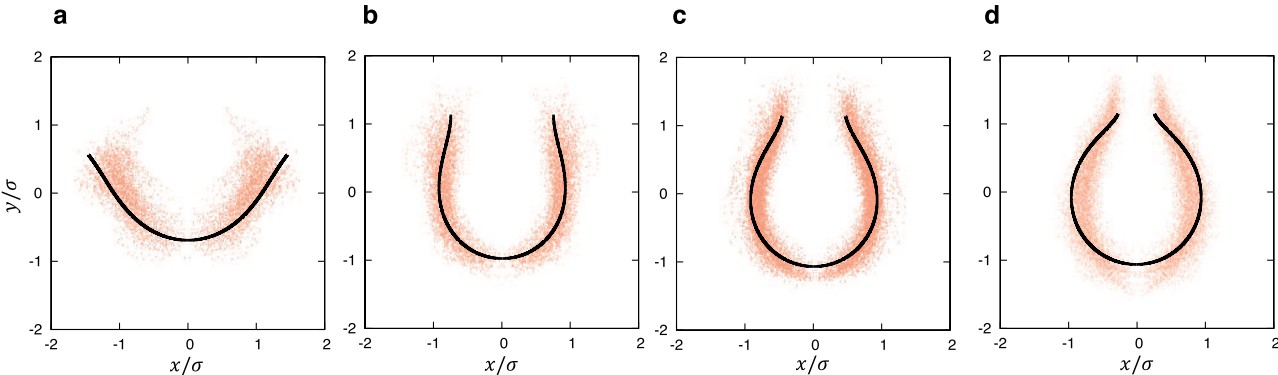

**Fig. 7 | Comparison of theoretical and experimental shapes.** The theoretically calculated shapes (black lines) were superimposed over the experimentally obtained point clouds (red) of very early-cups (**a**), early-cups (**b**), middle-cups (**c**), and late-cups (**d**) [using the same data as shown in Fig. 3e–h]. The theoretical results were normalized so that the mean of the point cloud representing each shape was zero and the variance was one, in the same way as the experimental results were normalized. The rim size $\sigma_{e_i}$ was fixed at $\sigma_{e_i} = 1.45$ (**a**), 0.75 (**b**), 0.48 (**c**), and 0.28 (**d**), corresponding to $l = 0.8$, 0.39, 0.24, and 0.13, respectively. Source data are provided as a Source Data file.

of phagophores is primarily determined by the relative size of the open rim to the total surface area without any external influences. We do not rule out the possibility that some internal forces (e.g., lipid asymmetry) or external forces (e.g., exerted by the endoplasmic reticulum or cytoskeletons) influence the phagophore morphology, but these effects would be relatively minor.

It should be noted that the shape of phagophores is scale-independent (Supplementary Fig. 1). Over a wide range of size scales, the theoretical model reproduced the experimentally observed morphology well. Generally, the bending energy of membranes [e.g., the outer and inner membranes of phagophores, the first term of Eq. (8)] is scale-independent[12], whereas that of the rim [i.e., the line energy, the second term of Eq. (8)] is scale (length)-dependent. Our result that the shape of phagophores is scale-independent suggests that the length of the rim is determined by a mechanism other than the simple bending energy model, probably by stabilization by curvature generators accumulating on the rim[6].

The rim length is considered to be constant on the time scale of mechanical relaxation as we assumed in this study, but it changes with phagophore growth on larger time scales. The change in the rim length could be explained by a change in the abundance of curvature generators at the rim that stabilizes the highly curved rim[6].

A recent cryo-electron tomography study in yeast shows that the intermembrane space between the inner and outer membranes is not constant, with slight swelling at the rim[23]. Rim swelling would change the bending elasticity, the local curvature, and the energy at the rim. However, the intermembrane distance is nearly constant outside the rim[23], consistent with the assumptions of our model. Although rim swelling was not explicitly considered in our model, it was considered as part of the line tension at the rim. It would be valuable to incorporate changes in intermembrane distance into the model and assess the effects of rim swelling on autophagosome shaping.

Recently, ATG2 was suggested to transfer lipids from the endoplasmic reticulum to autophagosomes[13–15]. The lipid transfer rate of ATG2 appears to be slow (approximately 0.017 lipid/s)[13], which is much slower than the lipid mechanical relaxation time on the order of milliseconds[6]. Thus, at each time point during autophagosome formation, the membrane shape should be equilibrated regardless of its expansion rate. However, another report indicates that the lipid transfer activity of ATG2 according to a kinetic model is much higher than experimental estimates[24] and may be comparable to the mechanical relaxation time of lipids, elevating the relevance of non-equilibrium effects. Non-equilibrium effects could reduce the elastic moduli of the membrane[25]. Therefore, it would be worth considering non-equilibrium effects during membrane elongation in future experiments.

The model also indicates that the surface energy [$\gamma_A A$ in Eq. (8)] is negative ($\gamma_A < 0$ in Fig. 6a–d), predicting that a larger membrane area is energetically more stable for the cup shapes. This stability could be one of the mechanisms underlying the directional lipid transfer from the endoplasmic reticulum to phagophores during phagophore growth.

## Methods

### Cell culture

Mouse embryonic fibroblasts (MEFs) were cultured in Dulbecco's modified Eagle's medium (DMEM: D6546, Sigma-Aldrich) supplemented with 10% fetal bovine serum (172012, Sigma-Aldrich), and 2 mM L-glutamine (25030-081, Gibco) in a 5% $CO_2$ incubator. For the starvation treatment, cells were washed twice and incubated in amino acid-free DMEM (048-33575, Wako Pure Chemical Industries) without serum. MEFs that stably express GFP-LC3 were generated previously[26].

### Three-dimensional electron microscopy of phagophores using array tomography

For chemical fixation, a glass base dish was coated with carbon using a vacuum evaporator (IB-29510VET, JEOL) and then coated with 0.1% gelatin and dried under UV irradiation[13]. Cells were cultured in the pretreated glass base dish for 2 days and starved for 30 min just before fixation. The cells were first fixed with 2.5% glutaraldehyde (G018/1, TAAB) in 0.1 M sodium cacodylate buffer, pH 7.4 (37237-35, Nacalai Tesque), overnight at 4 °C and then with 1% osmium tetroxide (3020-4, Nisshin EM) and 1.5% potassium ferrocyanide (161-03742, Wako) in 0.065 M sodium cacodylate buffer for 2 h at 4 °C. The fixed cells were washed five times with Milli-Q water, block-stained with 3% uranium acetate for 1 h, and dehydrated with an ethanol series. After a 1-h dehydration in 100% ethanol, the cells were embedded in Epon (EPON812, TAAB), polymerized at 40 °C for 12 h, and then polymerized at 60 °C for 48 h. Glass bases were removed from blocks by soaking them in liquid nitrogen, and then, the top surface of the blocks was trimmed to 100 ×100 mm using razor blades.

For rapid freeze fixation, we used the high-pressure freezing and freeze substitution method. A sapphire glass (EM ICE accessory, cat. number 10702766; Leica Microsystems, Wetzlar, Germany) was cleaned with ethanol, coated with carbon by using a vacuum evaporator (IB-29510VET), heat treated (60 °C for >48 h), coated with poly-L-lysin, and sterilized. Cells were cultured on the pretreated sapphire glass for 2 days and starved for 30–45 min before freeze fixation. The sapphire glass was then mounted between two aluminum carriers (EM ICE accessory, type A, cat. number 10660141). The space between the sapphire glass and carrier was filled with 3% percoll solution to

prevent the formation of ice crystals. After freezing in a high-pressure freezer (HPM010; Bal-Tec, Balzers, Liechtenstein), each sapphire disk was detached from an aluminum carrier in liquid nitrogen and quickly transferred to a tube for freeze substitution. In the tube, acetone was prepared containing 0.1% uranium acetate, 2% osmium tetroxide, and 3% water and then frozen with liquid nitrogen. Freeze substitution was carried out for 5 days at −80 °C, and then, the temperature was gradually raised, to −30 °C for 1 h and 40 min, to 4 °C for 1 h and 10 min, and to room temperature for 1 h. After being washed with dry acetone three times, the sapphire disks were collected and embedded in EPON812, with a gradually increasing EPON812 concentration, as follows: a 2:1 ratio of acetone:EPON812 overnight, a 1:1 ratio of acetone:EPON812 for 1 day, and 100% EPON 812 for 6 h. Then, the sapphire disks were placed in a silicon mold and polymerized. After polymerization, the sapphire disks were removed with a razor and by soaking in liquid nitrogen repeatedly.

To cut ultra-thin serial sections, we used a diamond knife with an ultra-jumbo boat (Ultra Jumbo 45 degree; Diatome, Hatfield, PA) mounted on an ultramicrotome (UC7; Leica). For chemical fixation, 25-nm-thick sections were cut, creating a ribbon of 130 serial sections, and for HPF fixation, 50-nm-thick sections were cut, creating a ribbon of 55 serial sections. The ribbons were collected onto a silicon wafer that was held and manipulated by an MM33 micromanipulator (Märzhäuser Wetzlar, Wetzlar, Germany). Electron microscopic images were acquired using a scanning electron microscope (JSM7900F; JEOL, Tokyo, Japan)[12].

To reduce the deformation effects caused by contact with other organelles, only phagophores that were not in contact with large cargos such as mitochondria were considered in the analysis. The small phagophores at very early stages were not included. In the 3D reconstruction, the scale factors in the stack direction were adjusted assuming axisymmetry of phagophores[17,18] in order to compensate for the compression of the structures during sample preparation (i.e., fixation and sectioning) for electron microscopy. For 3D reconstruction, the z-axis was roughly aligned in a large area (-10 × 10 mm) around autophagosomes (including other subcellular organelles) by StackerNEO (System in Frontier Inc., Tokyo, Japan), and serial images of each autophagosome were extracted. Next, the serial images were aligned by StackerNEO. For the segmentation of autophagosomes, their shapes were traced manually by Plot Digitizer.

## Fluorescence microscopy

MEFs stably expressing GFP-LC3B were subjected to live-cell fluorescence imaging using a SpinSR10 spinning-disk confocal microscope (Olympus) equipped with an ORCA-Flash 4.0 camera (Hamamatsu), a UPLAPO OHR 60× lens (NA 1.50, Olympus), a 3.2× optical zoom lens, and a SORA disk in place (Yokogawa). The microscope was operated with cellSens Dimension v2.3 (Olympus). During live-cell imaging, a dish was mounted in a chamber (INUB-ONI-F2, TOKAI HIT) to maintain the incubation conditions at 37 °C and 5% $CO_2$. Five slices with a thickness of 250 nm were acquired. The images were processed with cellSens Dimension software using the following settings: deconvolution with advanced maximum likelihood algorithm, 5× interaction, and adaptive PSF filter settings.

## Elastic bending energy and shape equations of membranes

For an axisymmetric geometry, the free energy of the membrane was given by

$$F = 2\pi \int_0^{s_1} \left[ \kappa_b x \left( \dot{\theta} + \frac{\sin\theta}{x} - J_0 \right)^2 + 2\kappa_G \dot{\theta}\sin\theta + 2\gamma_A x + \gamma_L \dot{x} \right] ds, \quad (12)$$

where $s$, $x$, $\theta$, and $J_0$ are the arc length, radial coordinate, tilt angle, and the spontaneous curvature, respectively (Fig. 5a)[16]; $s_1$ represents the location of the rim; $\kappa_b$ and $\kappa_G$ are the bending elastic moduli; and $\gamma_A$

and $\gamma_L$ are the Lagrange multipliers for the area $A$ and the rim length $L$, respectively. The free energy should be minimized with the constraints $\dot{x} = \cos\theta$, and thus the action $S = 2\pi \int_0^{s_1} \mathcal{L} ds$ was constructed with the Lagrangian

$$\mathcal{L} = \kappa_b x \left( \dot{\theta} + \frac{\sin\theta}{x} - J_0 \right)^2 + 2\kappa_G \dot{\theta}\sin\theta + 2\gamma_A x + \gamma_L \dot{x} + \gamma_x (\dot{x} - \cos\theta),$$
$$(13)$$

where $\gamma_x$ is the Lagrange multiplier for $\dot{x}$. Based on variational theory[16], the shape equation to determine the membrane shape with the minimum free energy is given by

$$\ddot{\theta} = -\frac{\tan\theta}{2}\dot{\theta}^2 - \frac{\dot{\theta}\cos\theta}{x} + \frac{\sin 2\theta}{2x^2} + \gamma_A\tan\theta + \frac{\tan\theta}{2}\left(\frac{\sin\theta}{x} - J_0\right)^2,$$
$$(14)$$

with the boundary conditions at the rim ($s = s_1$)

$$\left[ \dot{\theta} - J_0 + \left(1 + \frac{\kappa_G}{\kappa_b}\right)\frac{\sin\theta}{x} \right]_{s=s_1} = 0, \quad (15)$$

$$\left[ \left(1 + \frac{\kappa_G}{2\kappa_b}\right)\kappa_G\left(\frac{\sin\theta}{x}\right)^2 - \frac{\kappa_G}{\kappa_b}J_0\frac{\sin\theta}{x} - \gamma_A - \frac{\gamma_L}{2}\frac{\cos\theta}{x} \right]_{s=s_1} = 0. \quad (16)$$

These boundary conditions at the rim indicate that Gaussian curvature, line tension, and surface tension are obtained from the curvature at the rim. The initial conditions at $s = 0$ for $x$ and $\theta$ are $x(0) = \theta(0) = 0$.

By solving the shape equation with the boundary conditions, a stable shape is obtained for a membrane area $A = 4\pi \int_0^{s_1} x ds$ and rim radius $x_r = \int_0^{s_1}\cos\theta ds$, with the corresponding Lagrange multipliers $\gamma_A$ and $\gamma_L$, as shown in in Fig. 6a-d. The elastic moduli $(\gamma_A, \gamma_L, \kappa_G)$ affect the equilibrium shapes (Supplementary Fig. 7). As $\gamma_A$ increases and $\gamma_L$ decreases, rim radius $l$ decreases, and the membrane tends to close. In terms of free energy, in Eq. (9), $\gamma_A$ and $\gamma_L$ appear in the form of $F \sim \gamma_A \int x(s)ds + \gamma_L \int dx$. Negative $\gamma_A$ makes $x$ larger at each point $s$; as $x$ increases, $l$ becomes larger, and the membrane becomes flat. Meanwhile, $\gamma_L$ only affects $x$ at the boundary and also affects $\kappa_G$, which is determined by the boundary conditions as described in Eq. (15).

To compare theoretical expectations with the empirical data shown in Fig. 7, the computed shapes were normalized in the same way as in the experiment. That is, we constructed a point cloud of axisymmetric 3D shapes from the 2D shapes obtained in the calculation and normalized them using z-score normalization [see Eqs. (1–4)].

## Relation between line energy and bending energy at the rim

The elastic bending energy of the rim was determined by

$$F_r = \frac{\kappa_b}{2} \int (J_r - J_{r0})^2 dA_r \quad (17)$$

where $\kappa_b$ is the bending rigidity and $J_r$, $J_{r0}$, and $A_r$ are the total curvature, spontaneous curvature, and area, respectively, of the rim. The rim geometry was modeled as a part of a torus with an intermembrane radius $r$ (Fig. 5a). In the torus approximation, the total curvature and the surface element of the rim were respectively given by

$$J_r = \frac{1}{r} + \frac{\cos\varphi}{l + r\cos\varphi} \ , \ dA_r = 2\pi r(l + r\cos\varphi)d\varphi, \quad (18)$$

where $l$ is the rim radius[6]. The integration variable $\varphi$ spans the interval $[\theta - \pi/2, \theta + \pi/2]$, with the tilt angle $\theta$ at the rim (Fig. 5a). The intermembrane distance of phagophores was very small, and $l \gg r$ was

assumed. In this case, the total curvature and the surface element became $J_r = \frac{1}{r}$ and $dA_r = 2\pi r l d\varphi$, respectively, and then the bending energy became

$$F_r = \frac{\kappa_b}{2} \int_{\alpha-\pi/2}^{\alpha+\pi/2} \left(\frac{1}{r} - J_{r0}\right)^2 2\pi r l d\varphi = \kappa_b \pi^2 r \left(\frac{1}{r} - \bar{J_r}\right)^2 l = \gamma_L L, \quad (19)$$

with the rim length $L = 2\pi l$ and the line tension $\gamma_L = \frac{\kappa_b}{2}\pi r \left(\frac{1}{r} - \bar{J_r}\right)^2$. Therefore, the bending energy of the rim was converted into the line energy, which included the effects of the spontaneous curvature. The spontaneous curvature is induced by the curvature generating proteins, and the rim length is controlled by the abundance of curvature-generating proteins[6].

### Estimation of the Gaussian modulus with phagophore growth

According to the elastic bending energy model interpreted in terms of stresses[27], the membrane can develop a meridional bending moment, a force that resists bending, $M_m = \kappa_b J + \kappa_G J_p$; the first and second terms in this equation are associated with the total curvature, $J = J_m + J_p$, and the Gaussian curvature, $K = J_m J_p$, respectively (Fig. 5a), while $\kappa_b$ and $\kappa_G$ represent the bending and the Gaussian moduli, respectively, which characterize the elastic properties of the membrane. As shown in the electron micrographs (Fig. 3 and S1, S2), the phagophore had closely juxtaposed inner and outer membranes ($a$ and $b$ in Fig. 5a) with the same curvatures, and the rim can be approximated as a line ($c$ in Fig. 5a). The spontaneous curvature was assumed to be zero because the outer and inner membranes contain relatively few proteins[28]. Because the phagophore rim was not exposed to any external rotational moment, the total moment should be zero, that is, $M_m = 0$. From this moment balancing condition, the Gaussian modulus can be obtained as

$$\frac{\kappa_G}{\kappa_b} = -\frac{J}{J_p}. \quad (20)$$

Therefore, the Gaussian modulus, $\kappa_G$, is determined by fitting the experimental phagophore membrane shapes (Fig. 4), where the curvatures are obtained from Eq. (7).

The Gaussian modulus obtained by fitting the experimental phagophore membrane shapes with a polynomial function was $-(0.21 \pm 0.05)\kappa_b$, with no significant differences among the very early-, early-, middle- and late-cups (Fig. 5b). The deviation $\sigma_X$ defined in Eq. (1), which represents the spatial extent of the point clouds of each morphology (Fig. 3), was used as a unit of length (Supplementary Fig. 1). The obtained values of the phagophore membrane were within the reported range of biological membranes, namely, $-\kappa_b < \kappa_G < 0$[29,30].

### Reporting summary

Further information on research design is available in the Nature Portfolio Reporting Summary linked to this article.

## Data availability

Source data for all graphs of all figures are provided with the paper as Supplementary Information. Source data are provided with this paper.

## Code availability

The numerical code in the mathematical model is publicly available on GitHub at https://doi.org/10.5281/zenodo.10258267[31].

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

## Acknowledgements

We would like to thank Michael M. Kozlov and Masashi Tachikawa for useful discussions. This work was supported by a Grant-in-Aid for Scientific Research (KAKENHI) from JSPS (23K05715 to Y.S., 22H04635 to I. K-.H., and 22H04919 to N.M.) and Exploratory Research for Advanced Technology (ERATO; grant number JPMJER1702) from the Japan Science and Technology Agency (JST) (to N.M.). The numerical computations have been performed with the RIKEN supercomputer system (HOKUSAI).

## Author contributions

Conceptualization, Y.S. and N.M.; Methodology, Y.S.; Investigation, Y.S., S.T., C.S., and I.K-.H.; Formal analysis, Y.S.; Writing, Y.S., I.K-.H., and N.M.; Visualization, Y.S., S.T., C.S., and I.K-.H.; Funding acquisition, Y.S., I.K-.H., and N.M.; Resources, Y.S., S.T., C.S., and I.K-.H.; Supervision, Y.S. and N.M.

## Competing interests

The authors declare no competing interests.
