## [Peer Review File · Nature Communications]

Experimental determination and mathematical modeling of standard shapes of forming autophagosomesREVIEWER COMMENTS

Reviewer #1 (Remarks to the Author):

In their manuscript titled “Experimental determination and mathematical modeling of standard shapes of forming autophagosomes”, Yuji Sakai and co-workers have looked at the dynamic shape changes (morphological evolution) during autophagosomes formation. As per the current understanding, the pathway for this morphological evolution starts from a disk-shaped cisterna that undergoes bending into cup-shaped morphology. Eventually, this cup shape morphology ends up in double-bilayer spherical autophagosome. The authors’ work is presented in three parts. In the first part, they have done extensive image analyses of the 3-dimensional electron micrographs of 100+ phagophores and have classified the evolving shapes into four major categories (disk, early-cup, middle-cup and late-cup). The low-resolution noisy projected point clouds from the micrographs were fitted to axis-symmetrical polynomial function to arrive at average analytical shape functions. The analytical shape allows the authors to carry out characterization of the local mean and Gaussian curvatures as well quantify the rim geometry (radius). In the second part of their work, the authors develop an augmented Helfrich-like Hamiltonian and have tried to argue that the various non-transient shapes of the phagophores at different stages of the autophagy pathway can be faithfully modelled using classical continuum elastic theory. The Hamiltonian is written as a penalty functional on the mean and Gaussian curvatures and also on the overall membrane area and rim length. Finally, in the third part of their work, they compare the shapes derived by the image analyses process and the theoretical calculations and claim that the simple “elastic” and “equilibrium” mathematical model could capture the shapes and provide additional insights into the mechanics that stabilizes shape at a given stage of the phagosome evolution.

After going carefully through the paper, I find the work to be interesting but I have several concerns. Broadly speaking, changes in shape of phagophores is brought about by growing surface area of the membrane (induced by dynamic and constant inclusion of additional lipids to the the membrane) and the intermediate shapes are tightly regulated by curvature-sensing and curvature-inducing proteins that are strategically localized/positioned on the surface of the membrane. Also, the shape evolution is largely determined by the essential differences (both in lipid/protein composition and physical features) that exist in the outer membrane, inner membrane and the two flanking rim regions. As per my understanding, the gap between the inner and outer membrane is also a very important factor in the formation of the phagophore shapes. In both the image processing and the mathematical models, all these factors are assumed to be absent (even though the authors seem to be aware of them). I can understand some of the limitations that have led to such crude assumptions. For example, very low resolution of the electron micrograph does not allow the differentiation between inner and outer rim and (possibly) the bulb-like shape at the rim (due to merging small-vesicles). I find that this has oversimplified the nuances that exist in the phagophore shapes, which in turn has guided the mathematical model to be also very over-simplified. This factor become important especially in light of recently available higher-resolution cryo-ET images that show some remarkable features (at higher resolutions) unique to phagophores (Anna Bieber and co-workers, PNAS2022). I would like the authors to discuss the

work in light of the above restrictions they have imposed on their modelling. Ignoring important features (e.g. growing surface area (merging small vesicles), asymmetry in outer and inner membrane, gap length between the inner and outer membranes and positioning of proteins such as ATL etc) should have implications on the recapitulating the phagophore shapes faithfully, which doesn't seem to be the case in the current model. I do not think that "equilibrium at each step" assumption is a good one for these dynamic systems.

Specifically, I have the following concerns/comments that the authors may consider revisiting.

(i) "Although individual phagophores were not axisymmetric, superimposed point clouds appear axisymmetric around z-axis" – This seems like what is called "compensating/biased error". I think this has led to the oversimplified shapes. If the characterization was done on single image, would that reproduce a more characterise shape?

(ii) The four types of shapes were deduced from the rim length and the boundaries were chosen based on domain knowledge. If the authors had automated the process for the 100+ images and used the different available classification techniques, they may be able to generate clusters with similar shapes and then identify types from there. The populations of each cluster could also have informed the population/abundance of each type of shape giving additional insights into the autophagosome evolution process.

(iii) The rationale for developing the mathematical model seems to be off. The authors say that they developed the analytical model "to investigate whether the morphological features of the phagophores obtained by electron microscopy were reasonable". I think the model tells us more than that and the authors may revise/rephrase this accordingly.

(iv) The non-rim membrane is assumed to have zero curvature – how good is this assumption in the first term of the Hamiltonian. The flat cisterna has a favoured direction in which it bends (guided by asymmetry in outer and inner membrane).

(v) To me, it seems that the equilibrium shape is dominated by last term of the Hamiltonian (γL). Since all the modulus ($\kappa_G, \gamma_A, \gamma_L$) can be expressed in terms of κ_b , the authors should develop a phase diagram for shapes with different values of $\kappa_G, \gamma_A, \gamma_L$ as a function of varying κ_b . This exercise will bring about the role played by each term in the Hamiltonian towards shape changes.

(6) The material property of the phagophore keeps changing as the shape type changes (as per the model) – modulus values are different. Can the authors comment on this?

Reviewer #2 (Remarks to the Author):

In this study, Sakai and coworkers present room-temperature electron microscopy data and a derived mathematical model to describe the shape of autophagosomes in mouse embryonic fibroblasts and suggest that the characteristic shape of autophagosomes is based on elastic bending energy minimization. In essence, their shape would be determined by the size of the open rim with respect to the total area.

The manuscript is well-written and understandable. However, while I cannot comment on the mathematical derivation of the model, data quality, quantity, and model assumptions do not appear up to date.

More specifically:

1) Given the data quality bc. of chemical fixation, how are the authors sure they are looking at autophagosomes - especially at the earlier, less defined stages? Did they perform CLEM? Based on the low n-count (see also 2), could this not affect the data used for modeling? Since the structures were identified by hand, is there no chance of bias? (e.g., selecting only the most symmetrical cup shapes bc they resemble 'canonical' autophagosomes)

2) From 117 3D autophagosomes, this is split into four stages (disc, early cup, middle cup, late cup), which are divided into four sizes. Does this provide enough data points for each step?

3) In light of the recent study by Bieber et al. in native, unfixed cells using cryo-ET, is it not clear that the axial symmetry cannot be assumed? [the cited paper also goes back to 2009] Looking at both the EM and fluorescence images, is this not also clearly seen, at least for a significant fraction of the autophagosomes, in the authors' data?

Will this not significantly change both the data processing as well as modeling?

Finally, what is this work's implication and/or model-creating hypothesis? Bending energy minimization has long been discussed as a reason for (autophagosome) membrane closure. I agree that the present work is a necessary experimental extension to the authors' 2020 paper in iScience. However, it lacks either answering deep-rooted questions in the field, generating new critical hypothesis, or even highlighting new data acquired with cutting edge technology.

While I am sure that this work should be published, neither the analytical approach, the data, or novelty to the field in general make it a significant enough contribution to support its publication in Nature Communications.

Reviewer #3 (Remarks to the Author):

In the manuscript by Sakai Y. et al. the authors collect phagophore structures using array tomography. Using a point cloud function, the phagophores were segmented and the 3D information was extracted by z-score normalization and Delaunay triangulation. Similar shapes were observed by live cell fluorescence microscopy. The next step was to fit the point clouds with approximate curves. These structures were then used for mathematical modeling analysis. The structures obtained by mathematical modeling were finally compared with the experimentally determined structures. In most cases (except for the final stage), the intermediates were well represented by the mathematical model.

Mathematical modeling has been used in the past to describe features of phagophore biogenesis (PMID: 32891055, PMID: 22427874, PMID: 33473217). Here, the authors compare the model with experimental data and conclude that autophagosome formation is governed by elastic bending energy minimization. This confirms a similar conclusion described earlier (PMID: 22427874), now complemented by experimental data and predicted by mathematical modelling. In addition, the authors observed a catenoid intermediate structure that minimizes the surface area and reduces the bending energy of the rim. The data are very interesting and an important step toward understanding autophagosome formation. The model is still simple (the double membrane structure is simplified by a single curve) and a more complex description will be needed in the future to address the complex membrane transformations. However, this requires higher resolution EM analysis, such as cryo-electron tomography or block-face cryo-FIB/SEM imaging which goes far beyond the scope of the manuscript. I only suggest that the following points be addressed to improve the manuscript prior to publication.

1. I was very impressed by the fluorescence microscopy data showing the same stages as seen by electron tomography. Seeing the dynamic of the structural transformations during autophagy would be extremely desirable. Can the authors use live cell microscopy to follow individual events from start to end to see these morphological changes?
2. Recent studies using in situ cryo-electron tomography revealed that the phagophore rim in mammalian cells could have quite dramatic dilation phenotypes (PMID: 36917659, PMID: 36216808, PMID: 33154161). The authors should consider this in their statement in the discussion (page 10, line 294-296). Strong rim dilations will impact the overall total curvature influencing the maturation of the phagophore. This should be reflected in the discussion.
3. At page 7 line 167-169 the authors state that each intermediate is stable for several minutes. Is this really true? I am not convinced that the cited literature really shows this for each individual stage. The experiment suggested in point 1 would address this experimentally.
4. The authors state that they analyzed 100 structures. However, they should state how many structures they analyzed for each of the stages.
5. Title: There are two titles which is unusual the authors should decide on one of them.

6. Often the authors simplify the mathematical/physical terms as in the following case: “Note that $\gamma < 0$ was always negative (Fig. 6A-D), which indicates that membrane area growth decreased the energy of cup shapes”. What energy are the authors referring to?

7. The plural of radius is radii (page 8, line 206)

8. “The spontaneous curvature of the inner and outer membranes was assumed to be zero because the inner and outer phagophore membranes contain relatively few proteins”. This statement is rather weak since a lattice of proteins was suggested to span around the surface of the phagophore (PMID: 24485455). Moreover, during maturation, there will be an imbalance between proteins on the inside and outside. Since proteins on the outside are thought to be removed during biogenesis. Finally, certain regions, such as the phagophore rim, as the authors themselves state (PMID: 32891055), should be protein-rich.

9. Figure 2 E-H and 4 A-D, F-I lacks axis labeling. Same for 5,6, and 7

10. Calling the early stage a disc is misleading since none of the structures the authors describe is actually a disc in 3D – they all are bowl or cup-shaped.

Responses to the comments of Reviewer #1

In their manuscript titled “Experimental determination and mathematical modeling of standard shapes of forming autophagosomes”, Yuji Sakai and co-workers have looked at the dynamic shape changes (morphological evolution) during autophagosomes formation. As per the current understanding, the pathway for this morphological evolution starts from a disk-shaped cisterna that undergoes bending into cup-shaped morphology. Eventually, this cup shape morphology ends up in double-bilayer spherical autophagosome. The authors’ work is presented in three parts. In the first part, they have done extensive image analyses of the 3-dimensional electron micrographs of 100+ phagophores and have classified the evolving shapes into four major categories (disk, early-cup, middle-cup and late-cup). The low-resolution noisy projected point clouds from the micrographs were fitted to axis-symmetrical polynomial function to arrive at average analytical shape functions. The analytical shape allows the authors to carry out characterization of the local mean and Gaussian curvatures as well quantify the rim geometry (radius). In the second part of their work, the authors develop an augmented Helfrich-like Hamiltonian and have tried to argue that the various non-transient shapes of the phagophores at different stages of the autophagy pathway can be faithfully modelled using classical continuum elastic theory. The Hamiltonian is written as a penalty functional on the mean and Gaussian curvatures and also on the overall membrane area and rim length. Finally, in the third part of their work, they compare the shapes derived by the image analyses process and the theoretical calculations and claim that the simple “elastic” and “equilibrium” mathematical model could capture the shapes and provide additional insights into the mechanics that stabilizes shape at a given stage of the phagosome evolution.

After going carefully through the paper, I find the work to be interesting but I have several concerns. Broadly speaking, changes in shape of phagophores is brought about by growing surface area of the membrane (induced by dynamic and constant inclusion of additional lipids to the the membrane) and the intermediate shapes are tightly regulated by curvature-sensing and curvature-inducing proteins that are strategically localized/positioned on the surface of the membrane.

Also, the shape evolution is largely determined by the essential differences (both in lipid/protein composition and physical features) that exist in the outer membrane, inner membrane and the two flanking rim regions. As per my understanding, the gap between the inner and outer membrane is also a very important factor in the formation of the phagophore shapes. In both the image processing and the mathematical models, all these factors are assumed to be absent (even though the authors seem to be aware of them). I can understand some of the limitations that have led to such crude assumptions. For example, very low resolution of the electron micrograph does not allow the differentiation between inner and outer rim and (possibly) the bulb-like shape at the rim (due to merging small-vesicles). I find that this has over-simplified the nuances that exist in the phagophore shapes, which in turn has guided the mathematical model to be also very over-simplified. This factor become important especially in light of recently available higher-resolution cryo-ET images that show some remarkable features (at higher resolutions) unique to phagophores (Anna Bieber and co-workers, PNAS2022). I would like the authors to discuss the work in light of the above restrictions they have imposed on their modelling. Ignoring important features (e.g. growing surface area (merging small vesicles), asymmetry in outer and inner membrane, gap length between the inner and outer membranes and positioning of proteins such as ATL etc) should have implications on the recapitulating the phagophore shapes faithfully, which doesn't seem to be the case in the current model. I do not think that "equilibrium at each step" assumption is a good one for these dynamic systems.

RESPONSE:

We are pleased that Reviewer #1 was interested in our paper and provided positive feedback. As Reviewer #1 pointed out, there was not much discussion of the model assumptions and restrictions. Each of these concerns has been addressed in the revised Discussion section.

Growing surface area and non-equilibrium effects

As Reviewer #1 noted, our model assumes equilibrium conditions. Morphological changes during autophagosome formation (taking ~5–10 min) occur slowly. Indeed, our previous time-lapse imaging actually showed that the morphology at each stage can be stable for 0.5–1 min (a series of images from a published movie [Tsuboyama et al. 2016] are shown below).

Furthermore, the experimentally reported lipid transport rate of ATG2 is 0.017 lipid/s, which is sufficiently slower than the mechanical relaxation of the membrane on the order of milliseconds (Sakai et al., 2020). Therefore, our “equilibrium at each step” assumption is generally reasonable.

However, given that there is a paper suggesting that the lipid transfer rate of ATG2 may be higher than experimental estimates (Bülow and Hummer, 2020), we agree that it would be worth considering non-equilibrium effects.

We have added the following sentences to Lines 317–327.

“Recently, ATG2 was suggested to transfer lipids from the endoplasmic reticulum to autophagosomes (Maeda et al., 2019; Osawa et al., 2019; Valverde et al., 2019). The lipid transfer rate of ATG2 appears to be slow (approximately 0.017 lipid/s) (Maeda et al., 2019), which is much slower than the lipid mechanical relaxation time on the order of milliseconds (Sakai et al., 2020). Thus, at each time point during autophagosome formation, the membrane shape should be equilibrated regardless of its expansion rate. However, another report indicates that the lipid transfer activity of ATG2 according to a kinetic model is much higher than experimental estimates (Bülow and Hummer, 2020) and may be comparable to the mechanical relaxation time of lipids, elevating the relevance of non-equilibrium effects. Non-equilibrium effects could reduce the elastic moduli of the membrane (Almendro-Vedia, et al., 2017). Therefore, it would be worth considering non-equilibrium effects during membrane elongation in future experiments.”

Asymmetry between the outer and inner membrane and positioning of proteins such as ATL

Although our model did not consider the effects of spontaneous curvature owing to the asymmetry between the inner and outer membranes or the localization of proteins such as ATL, it would be interesting to discuss the effect of spontaneous curvature. Because this comment is related to Reviewer #1's Comment (iv) and Reviewer #3's Comment (8), we address this issue in our response to these specific comments below.

Gap length between the inner and outer membranes

It would be interesting to discuss the effect of changing the gap length between the inner and outer membranes, as it can indeed change the local curvature and energy. According to the Bieber et al. paper (PNAS 2022), the intermembrane space appears to be swollen within 50 nm of the rim tip, but remains constant for the most part as seen in Fig. 1E, F, Fig. 2B, and Fig. 4 of their paper.

We have added the following sentences to Lines 311–316.

“Rim swelling would change the bending elasticity, the local curvature, and the energy at the rim. However, the intermembrane distance is nearly constant outside the rim (Bieber et al., 2022), consistent with the assumptions of our model. Although rim swelling was not explicitly considered in our model, it was considered as part of the line tension at the rim. It would be valuable to incorporate changes in intermembrane distance into the model and assess the effects of rim swelling on autophagosome shaping.”

Specifically, I have the following concerns/comments that the authors may consider revisiting.

(i) “Although individual phagophores were not axisymmetric, Superimposed point clouds appear axisymmetric around z-axis” – This seems like what is called “compensating/biased error”. I think this has led to the oversimplified shapes. If the characterization was done on single image, would that reproduce a more characterise shape?

RESPONSE:

We thank this reviewer for pointing out the need for clarification. The purpose of this study is to extract morphological features common to phagophore membranes, rather than to examine features of individual morphologies. Analysis of many morphologies revealed that elongated cups and recurved rims are the common features of phagophores. These features also apply to individual morphologies, as shown in Figs. 3I-P, Fig. 4J, Video S1, and Supplementary Figs. 1–5, and thus, we do not think that this approach causes biased errors.

Given that the individual morphologies also show near axisymmetry as shown in Supplementary Figs. 1 and 4, we admit that the original sentence in Lines 119–121 (i.e., “Although individual phagophores were not axisymmetric, probably owing to deformation caused by intracellular fluctuations, the superimposed point clouds appeared to be axisymmetric around the z-axis”) was somewhat misleading. Supplementary Figs. 1 and 4 show individual point clouds projected onto a two-dimensional plane (considering only rim direction and distance from the axis), where axisymmetry is not assumed. The result shows that the projected point clouds are convergent and that the variance is small, indicating that axisymmetry is approximately valid for the individual shapes (if the axisymmetry is not good, the projected points will be scattered apart).

We rewrote the sentences in Lines 122–124 as follows.

“The superimposed point clouds showed less variation and appeared to be axisymmetric about the z-axis. This was also true for individual phagophores, although they were slightly deformed by intracellular fluctuations (Supplementary Figs. 1 and 4).”

(ii) The four types of shapes were deduced from the rim length and the boundaries were chosen based on domain knowledge. If the authors had automated the process for the 100+ images and used the different available classification techniques, they may be able to generate clusters with similar shapes and then identify types from there. The populations of each cluster could also have informed the population/abundance of each type of shape giving additional insights into the autophagosome evolution process.

RESPONSE:

We thank Reviewer #1 for this constructive comment. In the original manuscript, we classified phagophore morphology into four stages according to the rim length. It would certainly be interesting to examine whether the phagophore shapes can

be properly classified without prior knowledge. Using the k-means method without prior knowledge, which is commonly used in unsupervised machine learning classification, we classified all the point cloud images that we obtained (Supplementary Fig. 2). This classification generated four categories largely based on the rim length, although there was partial overlap between classes C and D (Supplementary Fig. 2E). We believe that this result validates our strategy for generating a mathematical model based on rim size.

Supplementary Figure 2. Unsupervised machine learning clustering of phagophore shapes.

Morphological images of phagophores were classified into four categories using the k-means method of unsupervised clustering learning. (A–D) Point clouds representing the morphology of each phagophore projected onto a two-dimensional plane belonging to each category are shown. (E) The rim length of phagophores in each category. Classes A–D correspond to the shapes of point clouds shown in panels A–D.

We also rewrote the sentences in Lines 109–112 as follows.

“... we first classified them into four categories using unsupervised machine learning clustering with the k-means method. The resulting four categories appeared to be primarily characterized by differences in rim size (Supplementary Fig. 2).”

Furthermore, since the initial shapes are not actually “disk”-like (pointed out by Reviewer 3), we renamed “disk” to “very early cup”.

(iii) The rationale for developing the mathematical model seems to be off. The authors say that they developed the analytical model “to investigate whether the morphological features of the phagophores obtained by electron microscopy were reasonable”. I think the model tells us more than that and the authors may revise/rephrase this accordingly.

RESPONSE:

We thank this reviewer for properly evaluating our model study. As Reviewer #1 mentioned, the stated rationale for the model study was somewhat ambiguous in our original version.

We restated the purpose of the model in Lines 170–173 as follows.

“Next, to investigate whether the morphological features of the phagophores obtained by electron microscopy at each stage could be spontaneously determined primarily according to the elastic model based on the physical properties of the membrane, we conducted a mathematical model analysis.”

(iv) The non-rim membrane is assumed to have zero curvature – how good is this assumption in the first term of the Hamiltonian. The flat cisterna has a favoured direction in which it bends (guided by asymmetry in outer and inner membrane).

RESPONSE:

In selective autophagy, the bending direction of flat cisternae can be affected by their cargo. However, in this analysis, only phagophores that were not in contact with large cargos, such as mitochondria, were considered.

We added the following sentence to the Methods section in Lines 383–385.

“To reduce the deformation effects caused by contact with other organelles, only phagophores that were not in contact with large cargos such as mitochondria were considered in the analysis.”

Differences in the protein and lipid composition of the inner and outer membranes can cause asymmetries, which is also noted by Reviewer #3 (Comment 8). Our experimental results show that phagophores adopt elongated cup-shaped morphologies. As shown in the new Supplementary Fig. 6, the spontaneous curvature makes the membrane morphology more spherical, which is inconsistent with our observation (elongated cups). Therefore, the magnitude of the spontaneous curvature and its effects on the shape would be small.

Supplementary Figure 6. Dependences of the shapes on spontaneous curvature.

Membrane shapes obtained based on the bending energy with Gaussian modulus $\kappa_G = -0.2\kappa_b$ and rim radii $l = 0.8$ (A), $l = 0.6$ (B), $l = 0.4$ (C), and $l = 0.2$ (D) for several values of spontaneous curvature J_0 . The unit of length was non-dimensionalized by the length $\sqrt{A/2\pi}$.

We added the following sentences to Lines 222–224.

“Spontaneous curvature made the membrane morphology more spherical (Supplementary Fig. 6). Our experimental results indicating that phagophores adopt elongated cup-shaped morphologies suggest that the spontaneous curvature of the membrane is not high.”

(v) To me, it seems that the equilibrium shape is dominated by last term of the Hamiltonian (γ_L). Since all the modulus ($\kappa_G, \gamma_A, \gamma_L$) can be expressed in terms of κ_b , the authors should develop a phase diagram for shapes with different values of $\kappa_G, \gamma_A, \gamma_L$ as a function of varying κ_b . This exercise will bring about the role played by each term in the Hamiltonian towards shape changes.

RESPONSE:

We are grateful for this constructive comment. It is valuable to investigate the role of each term of the Hamiltonian in the shape change. The dependence of membrane shape on the model parameters ($\gamma_A, \gamma_L, \kappa_G$) was investigated in the new Supplementary Fig. 7.

Supplementary Figure 7. Phase diagram of the membrane shapes on the membrane elastic moduli.

(A) Phase diagrams of the rim radius \tilde{l} were calculated on the $\tilde{\gamma}_A$ - $\tilde{\gamma}_L$ plane. The parameter sets corresponding to Figures 6A–D are represented as black dots. (B) The Gaussian modulus $\tilde{\kappa}_G$ at the corresponding position in panel A. The parameters were nondimensionalized as $\tilde{l} = l/\sqrt{A}$, $\tilde{\kappa}_G = \kappa_G/\kappa_b$, $\tilde{\gamma}_A = \gamma_A A/\kappa_b$, and $\tilde{\gamma}_L = \gamma_L \sqrt{A}/\kappa_b$, respectively. The white areas indicate regions in which the membrane cannot be realized in simple geometry, for example, because the sign of x or dz changes.

This figure shows that as $\tilde{\gamma}_A$ increases and $\tilde{\gamma}_L$ decreases, the rim radius \tilde{l} is smaller and the membrane tends to close. In terms of free energy, in Eq. (9), $\tilde{\gamma}_A$ and $\tilde{\gamma}_L$ appear in the form of $F \sim \tilde{\gamma}_A \int x(s)ds + \tilde{\gamma}_L \int dx$. Negative $\tilde{\gamma}_A$ makes x larger at each point s ; thus, \tilde{l} becomes larger, and the membrane becomes flat. Meanwhile, $\tilde{\gamma}_L$ only affects x at the boundary and also affects $\tilde{\kappa}_G$, which is determined by the boundary conditions, as described in Eq. (15).

We added the following sentences to Lines 233-235.

“In our model, the equilibrium shape depends on the membrane elastic moduli $(\gamma_A, \gamma_L, \kappa_G)$. The rim radius l increases with decreasing γ_A , and κ_G increases with γ_L (Supplementary Fig. 7).”

We also added the following sentences to the Methods section on Lines 424–430.

“The elastic moduli $(\gamma_A, \gamma_L, \kappa_G)$ affect the equilibrium shapes (Supplementary Fig. 7). As γ_A increases and γ_L decreases, rim radius l decreases, and the membrane tends to close. In terms of free energy, in Eq. (9), γ_A and γ_L appear in the form of $F \sim \gamma_A \int x(s)ds + \gamma_L \int dx$. Negative γ_A makes x larger at each point s ; as x increases, l becomes larger, and the membrane becomes flat.

Meanwhile, γ_L only affects x at the boundary and also affects κ_G , which is determined by the boundary conditions as described in Eq. (15).”

(6) The material property of the phagophore keeps changing as the shape type changes (as per the model) – modulus values are different. Can the authors comment on this?

RESPONSE:

The amounts and types of proteins and lipids in the phagophore membrane likely change as it grows and changes shape. For example, the amount of the lipid transport protein ATG2 increases with membrane growth and decreases as the rim radius decreases. Furthermore, the amount of ATG8 conjugated to membrane lipids increases. Changes in the composition of the proteins and lipids can alter the physical properties of the phagophore membrane.

We added the following sentences to Lines 232–237.

“As phagophore morphology changed, the elastic properties of the membrane also changed (Fig. 6A-D). In our model, the equilibrium shape depends on the membrane elastic moduli $(\gamma_A, \gamma_L, \kappa_G)$. The rim radius l increases with decreasing γ_A , and κ_G increases with γ_L (Supplementary Fig. 7). Therefore, as the membrane grows, changes in the lipid and protein composition of the phagophore membrane would modulate the physical properties of the membrane.”

Responses to the comments of Reviewer # 2

In this study, Sakai and coworkers present room-temperature electron microscopy data and a derived mathematical model to describe the shape of autophagosomes in mouse embryonic fibroblasts and suggest that the characteristic shape of autophagosomes is based on elastic bending energy minimization. In essence, their shape would be determined by the size of the open rim with respect to the total area.

The manuscript is well-written and understandable. However, while I cannot comment on the mathematical derivation of the model, data quality, quantity, and model assumptions do not appear up to date.

More specifically:

1) Given the data quality bc. of chemical fixation, how are the authors sure they are looking at autophagosomes - especially at the earlier, less defined stages? Did they perform CLEM? Based on the low n-count (see also 2), could this not affect the data used for modeling? Since the structures were identified by hand, is there no chance of bias? (e.g., selecting only the most symmetrical cup shapes bc they resemble 'canonical' autophagosomes)

RESPONSE:

Given that phagophores have a characteristic 3D morphology and their membranes exhibit high electron density (double membrane and high affinity to osmium (Ylä-Anttila et al. 2009), it is easy to identify these structures by 3D electron microscopy even without performing CLEM. As Reviewer #2 notes, it is true that very small structures at the very early stage might have been missed; however, these very early structures are not included in this study.

We added the following sentence to the main text (Lines 76–77).

“... based on their characteristic 3D morphology and strong contrast after osmium staining (Ylä-Anttila et al. 2009).”

Additionally, we added the following sentence to the Methods section (Line 385).

“The small phagophores at very early stages were not included.”

Ylä-Anttila P, Vihinen H, Jokitalo E, Eskelinen E-L. Monitoring autophagy by electron microscopy in Mammalian cells. *Methods Enzymol.* 2009;452:143-64. doi: 10.1016/S0076-6879(08)03610-0.

2) From 117 3D autophagosomes, this is split into four stages (disc, early cup, middle cup, late cup), which are divided into four sizes. Does this provide enough data points for each step?

RESPONSE:

As shown in superimposed views of multiple shapes at each stage (Fig. 3E-H), the location of the average point and its variance were sufficiently determined. Furthermore, as shown in Supplementary Fig. 2, an unsupervised machine learning method clearly classified the images into four categories largely based on the rim size, as we did manually in this study. Thus, we conclude that the sample size of each category was sufficient.

Supplementary Figure 2. Unsupervised machine learning clustering of phagophore shapes.

Morphological images of phagophores were classified into four categories using the k-means method of unsupervised clustering learning. (A–D) Point clouds representing the morphology of each phagophore projected onto a two-dimensional plane belonging to each category are shown. (E) The rim length of phagophores in each category. Classes A–D correspond to the shapes of point clouds shown in panels A–D.

We also rewrote the sentences in Lines 109–112 as follows.

“... we first classified them into four categories using unsupervised machine learning clustering with the k-means method. The resulting four categories appeared to be primarily characterized by differences in rim size (Supplementary Fig. 2).”

3) In light of the recent study by Bieber et al. in native, unfixed cells using cryo-ET, is it not clear that the axial symmetry cannot be assumed? [the cited paper also goes back to 2009] Looking at both the EM and fluorescence images, is this not also clearly seen, at least for a significant fraction of the autophagosomes, in the authors' data?

Will this not significantly change both the data processing as well as modeling?

RESPONSE:

The morphology of the phagophore in individual EM sections was certainly not symmetrical owing to intracellular fluctuations, as shown by Bieber et al. This was also true for our results shown in Fig. 3I-L. However, when the point clouds were superimposed, local fluctuations became less noticeable and the shapes appeared axisymmetric. In fact, we did not assume axisymmetry of the data in Fig. 3E-H, which shows the superimposed point clouds, where only the rim direction (i.e., mean edge vector) and distance from the axis were considered (in general, membranes can be represented in two dimensions). The result shows that the projected point clouds are convergent and that the variance is small, indicating that axisymmetry is approximately valid for individual shapes (if the axisymmetry is not good, the projected points will be scattered apart). The entire shape of individual structures also exhibit near axisymmetry, as shown in Supplementary Figs. 1 and 4.

4) Finally, what is this work's implication and/or model-creating hypothesis? Bending energy minimization has long been discussed as a reason for (autophagosome) membrane closure. I agree that the present work is a necessary experimental extension to the authors' 2020 paper in iScience. However, it lacks either answering deep-rooted questions in the field, generating new critical hypothesis, or even highlighting new data acquired with cutting edge technology.

While I am sure that this work should be published, neither the analytical approach, the data, or novelty to the field in general make it a significant enough contribution to support its publication in Nature Communications.

RESPONSE:

We respectfully disagree with this comment concerning the implications, novelty, and technology of our study. This is the first report determining the standard shape of forming autophagosomes (phagophores), rather than showing the physical mechanism of autophagosome closure. There has been no detailed

statistical and quantitative analysis of the morphology of growing phagophores (like our study dealing with more than 100 3D structures of entire phagophores). Our present results show that the shape of growing phagophores is not a simple partial sphere as previously thought, but instead has unique features, such as elongated cup shapes with an outwardly recurved rim, which can be quantitatively explained by our new model. These results are completely new and not merely an extension of our previous iScience paper. In most previous studies, the shape of phagophores was simply assumed to be part of a sphere or ellipsoid. This oversimplification has been one of the limitations of previous studies. In fact, in our previous iScience paper, we developed a model assuming phagophores take the form of a portion of an ellipsoid. With such limitations, it was not clear whether morphological changes of phagophores could truly be physically interpreted on the basis of bending energy.

Although the resolution of our methods is inferior to that of the high-resolution cryo-electron tomography conducted by Florian Wilfling's group (Bieber et al. PNAS 2022), we consider our methods to be more suitable to the aim of our study because they allow us to observe entire phagophores. It should be noted that only part of a phagophore and autophagosome, about 200 nm in thickness, can be observed by cryo-electron tomography, highlighting a key advantage of our method. We are confident that our results are completely new and not merely an extension of our previous iScience paper.

Responses to the comments of Reviewer # 3

In the manuscript by Sakai Y. et al. the authors collect phagophore structures using array tomography. Using a point cloud function, the phagophores were segmented and the 3D information was extracted by z-score normalization and Delaunay triangulation. Similar shapes were observed by live cell fluorescence microscopy. The next step was to fit the point clouds with approximate curves. These structures were then used for mathematical modeling analysis. The structures obtained by mathematical modeling were finally compared with the experimentally determined structures. In most cases (except for the final stage), the intermediates were well represented by the mathematical model.

Mathematical modeling has been used in the past to describe features of phagophore biogenesis (PMID: 32891055, PMID: 22427874, PMID: 33473217). Here, the authors compare the model with experimental data and conclude that autophagosome formation is governed by elastic bending energy minimization. This confirms a similar conclusion described earlier (PMID: 22427874), now complemented by experimental data and predicted by mathematical modelling. In addition, the authors observed a catenoid intermediate structure that minimizes the surface area and reduces the bending energy of the rim. The data are very interesting and an important step toward understanding autophagosome formation. The model is still simple (the double membrane structure is simplified by a single curve) and a more complex description will be needed in the future to address the complex membrane transformations. However, this requires higher resolution EM analysis, such as cryo-electron tomography or block-face cryo-FIB/SEM imaging which goes far beyond the scope of the manuscript. I only suggest that the following points be addressed to improve the manuscript prior to publication.

1. I was very impressed by the fluorescence microscopy data showing the same stages as seen by electron tomography. Seeing the dynamic of the structural transformations during autophagy would be extremely desirable. Can the authors use live cell microscopy to follow individual events from start to end to see these morphological changes?

RESPONSE:

We have previously traced these morphological changes from start to end, using live-cell fluorescence microscopy (e.g., Fig. 1A in the paper by Tsuboyama, Koyama-Honda, et al., Science 2016). Time-lapse images at 10-second intervals are shown below (scale bar, 2 μm).

To avoid duplicate publication, we show these images only here in this letter and did not include them in the present paper.

2. Recent studies using in situ cryo-electron tomography revealed that the phagophore rim in mammalian cells could have quite dramatic dilation phenotypes (PMID: 36917659, PMID: 36216808, PMID: 33154161). The authors should consider this in their statement in the discussion (page 10, line 294-296). Strong rim dilations will impact the overall total curvature influencing the maturation of the phagophore. This should be reflected in the discussion.

RESPONSE:

It would be interesting to discuss the effect of changing the gap length between the inner and outer membranes, as it can indeed change the local curvature and energy. According to the Bieber et al. paper (PNAS 2022), the intermembrane space appears to be swollen within 50 nm of the rim tip, but remains constant for the most part as seen in Fig. 1E, F, Fig. 2B, and Fig. 4 of their paper.

We have added the following sentences to Lines 311–316.

“Rim swelling would change the bending elasticity, the local curvature, and the energy at the rim. However, the intermembrane distance is nearly constant outside the rim (Bieber et al., 2022), consistent with the assumptions of our model.

Although rim swelling was not explicitly considered in our model, it was considered as part of the line tension at the rim. It would be valuable to incorporate changes in intermembrane distance into the model and assess the effects of rim swelling on autophagosome shaping.”

3. At page 7 line 167-169 the authors state that each intermediate is stable for several minutes. Is this really true? I am not convinced that the cited literature really shows this for each individual stage. The experiment suggested in point 1 would address this experimentally.

RESPONSE:

We thank this reviewer for pointing this out. As shown in the figure in response to comment #1 of Reviewer #3, the morphology of each stage appears to be stable for 0.5–1 min. Thus, we have changed the phrase “several minutes” to “tens of seconds to a minute” (Line 174).

4. The authors state that they analyzed 100 structures. However, they should state how many structures they analyzed for each of the stages.

RESPONSE:

Although we have mentioned it in the caption to Fig. 3, we have added the following phrase to the text on Lines 117–118: “with 15, 16, 24, and 45 observations, respectively.”

5. Title: There are two titles which is unusual the authors should decide on one of them.

RESPONSE:

We apologize for the confusion. The second title was intended to be the short title. We have now clarified this accordingly.

6. Often the authors simplify the mathematical/physical terms as in the following case: “Note that $\gamma <$ was always negative (Fig. 6A-D), which indicates that membrane area growth decreased the energy of cup shapes”. What energy are the authors referring to?

RESPONSE:

We thank Reviewer #3 for noting this issue. We think that a numbering error in the equation was misleading. Eq. (10) in the previous manuscript should have

referred instead to Eq. (8). Given that γ_A was negative, the surface energy was also negative, and the free energy decreased as the membrane area increased.

We also rewrote the following sentence in Lines 237–239.

“Note that γ_A was always negative (Fig. 6A-D), which indicates that the surface energy, the second term of Eq. (8), was negative, and thus, membrane area growth decreased the energy of cup shapes.”

7. The plural of radius is radii (page 8, line 206)

RESPONSE:

We have corrected this accordingly.

8. “The spontaneous curvature of the inner and outer membranes was assumed to be zero because the inner and outer phagophore membranes contain relatively few proteins”. This statement is rather weak since a lattice of proteins was suggested to span around the surface of the phagophore (PMID: 24485455). Moreover, during maturation, there will be an imbalance between proteins on the inside and outside. Since proteins on the outside are thought to be removed during biogenesis. Finally, certain regions, such as the phagophore rim, as the authors themselves state (PMID: 32891055), should be protein-rich.

RESPONSE:

Any proteins that are present on the phagophore membrane surface are incorporated into our model as the spontaneous curvature of the membrane owing to the asymmetry of the protein and lipid composition between the inner and outer membranes. It would be valuable to consider the spontaneous curvature, which is also noted by Reviewer #1 (iv). We investigated the effect of the spontaneous curvature on the morphology of phagophores. Our experimental results show that phagophores adopt elongated cup-shaped morphologies. As shown in the new Supplementary Fig. 6, the spontaneous curvature makes the membrane morphology more spherical, which is inconsistent with our observation (elongated cups). Therefore, the magnitude of the spontaneous curvature and its effects on the shape would be small.

Supplementary Figure 6. Dependences of the shapes on spontaneous curvature.

Membrane shapes obtained based on the bending energy with Gaussian modulus $\kappa_G = -0.2\kappa_b$ and rim radii $l = 0.8$ (A), $l = 0.6$ (B), $l = 0.4$ (C), and $l = 0.2$ (D) for several values of spontaneous curvature J_0 . The unit of length was non-dimensionalized by the length $\sqrt{A/2\pi}$.

We added the following sentences to Lines 222–224.

“Spontaneous curvature made the membrane morphology more spherical (Supplementary Fig. 6). Our experimental results indicating that phagophores adopt elongated cup-shaped morphologies suggest that the spontaneous curvature of the membrane is not high.”

9. Figure 2 E-H and 4 A-D, F-I lacks axis labeling. Same for 5,6, and 7

RESPONSE:

We have added the appropriate labels to the figures.

10. Calling the early stage a disc is misleading since none of the structures the authors describe is actually a disc in 3D – they all are bowl or cup-shaped.

RESPONSE:

We renamed “disk” to “very early cup”.

REVIEWERS' COMMENTS

Reviewer #1 (Remarks to the Author):

Good from my end. Thanks.

Reviewer #3 (Remarks to the Author):

My concerns were all addressed. I congratulate the authors for this beautiful work.